# Learning Representations from Pre-synaptic Glomerular Responses for Odor Classification

## Abstract

Advances in neural sensing technology now make it possible to observe presynaptic responses in the olfactory bulb with high spatial and temporal resolution. In this paper, we approach olfaction (the sense of smell) from a Machine Learning perspective, focusing on how odor identity can be decoded from neural representations at the first synaptic stage of the olfactory system. Drawing distinctions to color vision, we argue that smell presents unique measurement challenges, including the complexity of stimuli, the high dimensionality of the sensory apparatus, as well as what constitutes ground truth. In the face of these challenges, we argue for the centrality of odorant-receptor interactions in developing a theory of olfaction. Such a theory is likely to find widespread healthcare applications in disease diagnostics, enhance our understanding of smell, and in the longer-term can help us understand how it relates to other senses and language. As an initial use case, we show that machine learning models can learn meaningful representations from calcium imaging of glomerular activations, enabling accurate odorant classification and revealing that pre-synaptic responses at the first olfactory synapse encode rich, discriminative information about odor identity. Additionally, we release 'oMNIST'[1] — a standardized dataset of glomerular responses for public use—designed to catalyze research in classification, representation learning, cross-animal glomeruli alignment and generalization in olfaction.

***Keywords***: *Olfaction, Smell, Odorants, Ligands, Olfactory Bulb, Glomeruli, Physico-Chemical properties*

## 1 Introduction

Smell is arguably the most primal and yet least understood of the senses. It has been key to the survival and fitness of a large number of species, for identifying or locating food, sensing danger, driving social behaviors, tracking and navigation, and much more. Smell provides vital sensory data to the brain, but remains poorly understood as a sense for a number of reasons. At the moment, we are unable to explain the relationship between the physical and perceptual properties of odors.

In contrast, human vision is relatively well understood. Objects have properties such as shape, size, and color. We tend to agree easily whether we are seeing a banana or a house, although we might initially mistake a Chihuahua for a muffin. In other words, we largely agree on what we are seeing based on shape, size, and color.

Color, which is well understood, provides an interesting reference for understanding smell. The theory of color vision is expressed in terms of *trichromacy*, *linearity*, and *opponency* — the three-dimensionality of perceptual color space, and the independence and nature of those dimensions, respectively. These concepts were formulated and experimentally validated in the late nineteenth century. As early as 1922, with the publication of the Optical Society of America's colorimetry report, clear definitions of terms were established, both psychological and physical (Troland, 1922). The publication of physical standards and experimental methods allowed for the characterization of

---

[1]Dataset link omitted for double-blind review. It will be released after review is completed. Processed data alongwith the code is available at  Github Repository

observers, culminating with the International Commission on Illumination's formalization of both colorspace and the 'standard observer' in 1931.

We begin by showing why such advances were possible for color vision early on in modern science, while they remain challenging for olfaction research today. We do this by comparing the two sensory modalities at the levels of their physical stimuli, neural systems, and the nature of perceptual experiences that observers can report, as summarized in Figure 1. We argue that the complexity of olfaction, arising from both stimulus structure and neural responses, demands a joint contribution of neuroscience and machine learning—where biological experiments provide the data and constraints, and representation learning methods uncover structure and principles of odor coding. Towards this end, we make available a standard dataset consisting of odors and the neural signatures they generate across transgenic mice. The dataset will be augmented along both dimensions – the number of odorants and their associated responses, as additional data is recorded. We hope to catalyze olfaction research in Neuroscience and Machine Learning, similar to how MNIST led to rapid advances in machine vision.

Like MNIST, where each digit has multiple renditions that can be used as training data, each odor in our database has multiple neural responses associated with it, along with metadata about each animal. Researchers can build on this data platform in many ways, from data handling to predictive modeling. On the data handling side, for example, we might expect innovation in methods for compressing or removing noise from the neural time series data. Indeed, we illustrate the use of two such methods and their downstream impacts on odor prediction based on simple machine learning based models. On the predictive modeling front, we demonstrate the use of AI models for prediction that aim to learn the distribution of odor space. In the longer term, we aim to develop generative models that bridge molecular features, neural representations, and linguistic descriptions of odors, enabling the synthesis of novel odors.

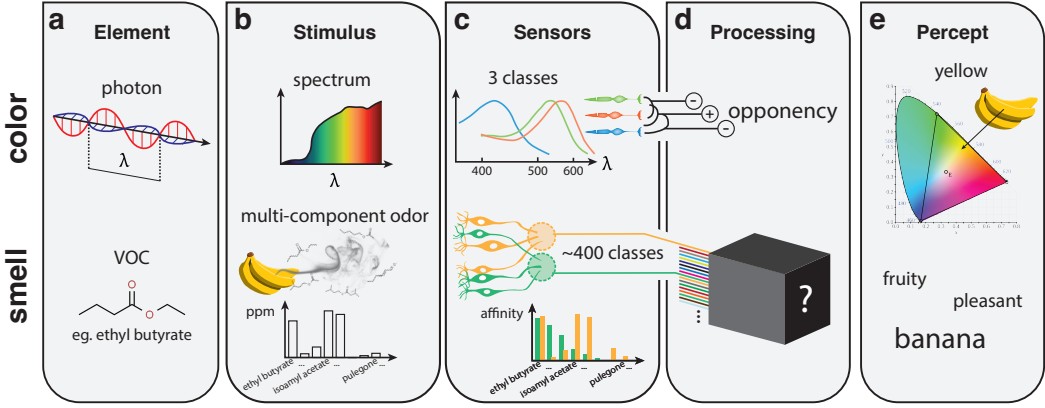

Figure 1: **A comparison of color vision (top) and olfaction (bottom).** a) The elements of the physical stimulus for color are photons, which can be defined by the single parameter of wavelength. The elements of odors are volatile organic compounds (VOCs), which have diverse molecular structures. b) Elements combine into spectra and multi-component odors. c) Just three sensor classes support trichromatic human color vision, with their well-defined spectral response profiles. In contrast humans have approximately 400 classes of olfactory receptor, with mostly undiscovered chemical receptive fields. Olfactory sensory neurons expressing the same receptor type (indicated by color) project to the same structure in the olfactory bulb, called a glomerulus. d) Human color vision is three-dimensional due to the way the three receptor type channels are compared with opponent processing. It is unclear how the information from olfactory receptor channels is compared or transformed, leaving it currently as a 'black box'. e) Color sensations can be described with semantic labels, but can also be located in the isoluminant plane, which enables accurate predictions for operations in color space such as mixing. Smells can be described in many ways, including objects of origin (banana), broader categories (fruity), pleasantness, and other descriptors, and mixing is poorly understood.

## 2 A COMPARATIVE TOUR THROUGH COLOR VISION AND OLFACTION

### 2.1 ODORS RUN THROUGH A LARGER GAMUT THAN COLORS

We begin by describing olfaction relative to color vision. Such a comparison enables us to highlight the unique aspects of olfaction and motivate our approach and the development of a standard benchmark dataset.

The physical stimulus for color vision are spectra of light, the elements of which are photons whose wavelengths fall in the visible range, namely 380-750 nm (Fig. 1.a). Each stimulus hitting the retina has a corresponding spectrum, which describes its amplitude as a function of wavelength (Fig. 1.b). While the spectrum of visible light is continuous, such that photons could have any of an infinite number of possible wavelengths, in practice wavelengths are sampled with a certain resolution. For example, if a spectrum is be reported with a wavelength resolution of 10 nm, for the visible range between 380 and 750 nm, it will be measured along 37 dimensions organized along the single linear dimension of wavelength (i.e. 380-389 nm, 390-399 nm, ...740-749 nm). A spectrum then corresponds to a point in 37-dimensional space. Spectra reaching the retina are generally determined by the emissive properties of light sources, as well as the reflective and transmissive properties of objects.

In contrast, the elements of physical stimuli for smell are volatile organic compounds, or VOCs (Fig. 1.a). Unlike photons, VOCs cannot be described by a single continuous dimension such as wavelength. Instead, the space of VOC molecular structures is discrete and highly highly multidimensional, with no obvious way of relating the myriad functional groups and moieties thaht exist in practically infinite combinations and permutations. For example, we can organize the esters ethyl propionate, ethyl butyrate, and ethyl pentanoate by the increasing carbon chain length of their parent acids, but other dimensions are required to relate branching (e.g. ethyl isovalerate) or other functional groups (e.g. ethyl benzoate). In summary, while photons can be defined by their nanometer wavelength, VOCs require the full IUPAC nomenclature, with all its prefixes, suffixes, and infixes, which indicates the complexity in naming and categorizing complex molecular structures.

The multidimensionality of VOC space presents an immediate challenge in how to select stimuli for research in olfaction. Unless a low-dimensional structure can be found in structure space, paving the way for a smaller number of 'olfactory primaries,' olfaction experiments may require the use of thousands of monomolecular odorants, which must be synthesized, purified, or purchased at significant financial and operational cost. Understanding how this large space of stimuli is sensed and how to optimize its exploration are questions that are particularly suitable for data science. At the longer term, we should understand multi-component "natural" odors that are created from the dynamic equilibria of multiple metabolic pathways, which can consist of hundreds or even thousands of VOCs (Fig. 1.b). Using more naturalistic smells may be critical for investigating natural odor abilities, for example the communication of social signals, analogous to higher-order visual processes such as the perception of faces.

### 2.2 OLFACTION IS A BLACK BOX, NOT A PRISM

Our understanding of color vision has emerged over centuries, based a progressive understanding of light and our vision system (Grassmann, 1854; Maxwell & Niven, 2011), which includes channels tuned for carrying color information as well as the concept of "opposing" colors. The discovery of three distinct and independent channels carrying color information corresponding to the three classes of cone photoreceptors in the retina (Fig. 1.c) and how they combine has been key to understanding the structure of color space.

Because of the relatively simple input-layer architecture of the human color vision system comprising three independent channels, we can liken it to a kind of coarse prism. Any incoming stimulus, whether monochromatic or made up of multiple wavelengths, gets split into three channels corresponding to the short-preferring (S), middle-preferring (M), and long-preferring (L) photoreceptor types, whose spectral response profiles are generally themselves organized along the single dimension of wavelength. The three-dimensionality of colorspace is a consequence of these three input channels, their independence, and the visual system's subsequent comparing of these channels through "opponent" processing (Fig. 1.d).

In contrast, olfactory receptors (ORs) form the largest family of G-protein coupled receptors in the mammalian genome (Gaillard et al., 2004), accounting for approximately 3% of the genome, with hundreds of channel types. Humans have roughly 400 receptor types, while the mouse olfactory system has roughly 1,100 types of receptors. And rather than a lock-and-key situation where ligands and receptors are exclusively matched, ORs exhibit broad tuning. A single receptor type will have affinities for a large number of odorant molecules via its promiscuous binding site, and a single odorant will bind to a large number of receptors to varying degrees. Each OR type therefore gives rise to an input channel with a broad receptive field spread throughout high-dimensional VOC structure space.

The olfaction case is further complicated by the potential for interactions between odorants at the level of receptor-ligand kinetics. While in vision photons linearly sum at the photoreceptor level, odorants may exhibit inhibitory, antagonistic and synergistic interactions when binding olfactory receptors (Inagaki et al., 2020). Although it is likely such non-linear effects are more the exception than the rule, we cannot safely assume the same kind of vector space properties as Maxwell wrote to describe color, such as linear addition and interpolation to predict mixtures. Instead, the neural response is a complex time series activation of ORs.

These features, namely the large number of sensor types, the complex receptive fields of individual sensors, and the potential of interactions during olfactory signal transduction, currently render olfaction as more of a black box than the relatively transparent prism of color vision. They also prevent us from characterizing the olfactory system's input layer properties as exhaustively as we have done for color, in the form of comprehensive receptor sensitivity functions (Fig. 1.c). There are simply too many receptor types for which we would have to measure affinity for too many (practically infinite!) odorants at various levels of concentration, let alone their potential interactive combinations.

## 2.3 THE DIVERSE OBSERVER AND GROUND TRUTH

We complete the comparative tour of color and olfaction by considering the diversity of the population of human observers. Color vision is remarkably well-conserved among observers, who generally are in high agreement when making perceptual judgments. This is because the majority of the population are 'normal' trichromats expressing three photoreceptor types, one each from the L, M and S opsin genes. Providing participants do not have some form of congenital color vision deficiency (approx. 8% of males, 1% of females), the results from color matching experiments and other studies show minimal individual differences. This allows us to define a 'standard observer' model of how the majority of people experience color.

On the other hand, olfactory tasks often lead to disagreement among observers, in how they determine both thresholds (Stevens et al., 1988) and quality (Mainland et al., 2013). There are numerous factors that contribute to differential sensitivities of the olfactory system, including biological factors such as age, gender, and genetic ancestry, as well as factors relating to experience, lifestyle and culture. Recently it has also been shown that the particular subset of OR genes expressed, which varies wildly across individuals, also contributes to disagreements observers make about the intensity, similarity, or pleasantness of olfactory stimuli (Trimmer et al., 2017). Specific anosmias, the olfactory equivalent of color vision deficiencies, where observers are insensitive to a particular odor, are widely prevalent and more of a rule than an exception (Croy et al., 2016). For olfaction, we should therefore expect a wide distribution of diverse observers properties, with a corresponding high degree of linguistic variance associated with odor descriptions.

Despite the baked-in variance that comes with linguistic descriptors, meaningful progress is being made by relating semantic label data to the physico-chemical features of odorants. A number of models have been developed to predict semantic labels from chemical structure, with the most recent advance employing a graph neural network that takes only the molecule's atoms and chemical bonds as input to predict its linguistic descriptors taken from the Good Scents and Leffingwell & Associates food and fragrance databases, for over five thousand odorants (Lee et al., 2023). The model was able to learn an optimized embedding function that transformed the graph of a molecule's atoms and bonds to a 256 dimensional embedding, which the authors call a principal odor map (POM). This POM is then further transformed to a read out semantic descriptors. The final trained model achieved an area under the receiver operating characteristic curve of 0.89 for a held out test set of 20% of the odors, a slight improvement over a random forest model trained to predict labels the

Mordred physico-chemical descriptor dataset, which had an AUROC of 0.85 (Sánchez-Lengeling et al., 2019). It should be noted however that both the precision (whether label predictions were correct or incorrect) and recall (whether odorant labels were predicted or missed) of this model were below 0.4, indicating the limits of language as ground truth.

While we would ultimately like to reconnect smell to perception and language, we use biology for two reasons. Fundamentally, neural responses provide an objective and robust ground truth for olfaction and may provide the foundation for a causal theory of olfaction. This is the approach we have taken towards building an olfaction database. Using a standard strain of mice, we record their neural signatures for odors multiple times and at various concentrations. Using animals from of a single genetic line limits variability at the level of OR expression. Until such time as we develop a digital nose that has an equivalent degree of sensory ability as a biological one, we will need to rely on biological sensors to provide us with the ground truth required to build predictive models.

## 3 FROM ODORS TO ODOR SPACE - LEARNING REPRESENTATIONS

The abundance of digital images and language data has spurred rapid progress in AI for vision and language, while olfaction has lagged behind because of its complexity and the difficulty of collecting high-quality data. Recent advances in optical methods now make it possible to record detailed odor-evoked activity from the olfactory bulb of mice, providing a foundation for data-driven representation learning. By observing odor responses at the first synapse, we capture the earliest neural codes before they are transformed by higher brain regions.

Why measure responses in mice, and not directly in human observers? It would be ideal to record the activity of the human olfactory system, which would allow us to correlate perceptual judgments such as linguistic labels with their neural representations. However, current noninvasive brain recording modalities, such as fMRI, are severely limited in both temporal and spatial resolution, such that the fine-grained details of neural representations are mostly unobservable. A recent study using fMRI found odor-specific neural activity in the aorbitofrontal cortex to be predictive of linguistic descriptors (Sagar et al., 2023), but the authors did not analyze activity in the olfactory bulb, the first brain region where odor information is delivered from the nose, likely due to the insufficient spatial resolution of fMRI.

Another option would be to grow cell lines, genetically modified to express human olfactory receptors, cultured to grow *in vitro*. Such cell lines can then be evaluated with ligand binding assays to measure odorant-receptor interactions. While this approach has been achieved experimentally, *in vitro* OSNs demonstrate far reduced sensitivity to those *in vivo*. With mice, a cranial window can be surgically implanted above the olfactory bulb to gain optical access to glomeruli, the neuropil structures in the olfactory bulb where axons of olfactory sensory neurons expressing the same class of receptor conveniently aggregate (as shown in Figure 1.c). Using mouse lines which have been engineered to express genetically encoded calcium indicators, we can record odor-evoked spatial-temporal glomerular responses using a camera, seeing glomeruli literally 'light up' as a function of neural activity. Another benefit of imaging the olfactory bulb is that glomeruli for the same receptor type exhibit stereotyped spatial locations across animals, allowing for a good (but not exact) alignment of data collected in multiple mice (Soucy et al., 2009). Observing mice also enables us to build a model that is potentially transferable to humans, due to the highly conserved nature of the mammalian olfactory system with respect to both receptor subfamilies (Godfrey et al., 2016) and system architecture (with the exception of the vomeronasal organ and accessory olfactory bulb (Lane et al., 2020).

A curated database of these odor-evoked responses provides a foundation for representation learning in olfaction research. For example, some models may explore the relationship between the physico-chemical features of individual VOCs and their corresponding representations in the olfactory system, while others might uncover principles that govern mixing in multi-component odors, or how perceptual properties arise from olfactory stimulii. In all these cases, data that inform on stimulus-receptor relationships *in situ* within the mammalian olfactory system will provide a lot of value, as ultimately it is these relationships that determine many aspects of olfaction. We present results of odor identification from videos of glomerular activation patterns. Our end-to-end pipeline ingests raw activation frames, denoises them, and learns representations that are predictive of odor identity. Importantly, we show how odors map onto unique parts of the olfactory bulb.

# 4 EXPERIMENTS AND RESULTS

## 4.1 DATASET

The neural data were collected from mice expressing calcium indicators, GCaMP6f (Chen et al., 2013) in olfactory sensory neurons (OSNs). This biosensor, GCaMP6f is designed to fluoresce in response to changes in intracellular calcium concentrations, thus serving as an indicator for neuronal activity which is recorded using a camera.

Our database structure consists of neural activation patterns of two mice that are exposed to mono-molecular odorants. For each mice and an odorant, there are eleven trial videos/ number of data points available for creating the train and test datasets. As an initial proof of concept, we have chosen 35 odorants that elicit strong activation of glomeruli in the dorsal olfactory bulb, and cover several important classes of odorants that include aldehydes, esters, ketones, and acids. Such diversity is essential for achieving a comprehensive perspective on glomerular activation, where we can isolate how the different chemical classes interact distinctively with specific types of olfactory receptors. We are continually expanding the dataset with additional odors and animals, and make the raw data publicly available to facilitate fair comparison, benchmarking, and reproducible research in olfaction.

## 4.2 BUILDING DATA INGESTION PIPELINE

For each odorant–mouse pair, glomerular activity was recorded as a video sequence (Appendix A; Figure 6). Each recording was converted into a stack of image frames sampled at the frame rate of the camera. Since raw videos are inherently noisy, we applied preprocessing steps to extract discriminative neural signals while suppressing background structure. In particular, widefield imaging captures strong anatomical signals such as vasculature, which dominate the images but do not contribute to odor identity. Our objective is to isolate the foreground patterns of glomerular activation, which carry the relevant stimulus-specific information.

Appendix A; Figure 7 shows a schematic of this process. The left panel shows a raw image frame containing both background and glomerular activity, while the middle panel shows the background alone. The similarity of these two frames highlights the difficulty to discern activations directly from the raw data. Subtracting the background from the raw frame yields the right panel, which reveals the foreground activation pattern for the odorant.

To further condense the temporal information, each video sequence of 320 frames was projected into a single image by computing the maximum pixel intensity (MPI) across time for each pixel (Appendix A; Figure 8). The resulting MPI images were then enhanced using Median Filtering (Huang et al., 1979) followed by anisotropic (Perona–Malik) diffusion (Weickert, 1996; IEE). This procedure reduces noise and sharpens region boundaries, improving the signal in regions of interest (ROIs) (Appendix A; Figure 9). The final MPI images serve as standardized inputs for our machine learning model. Each image represents a trial-specific neural signature of glomerular activation, from which models learn spatial representations that enable odor classification.

## 4.3 RESULTS: LEARNING NEURAL REPRESENTATIONS FOR ODOR CLASSIFICATION

We evaluated our approach by training and testing on aggregated, denoised MPI images. The dataset was split into train and test sets, with training and testing images corresponding to different mice. This allowed us to assess whether our model trained on one subject can generalize at the level of the compressed spatial representations of ROIs across subjects. We expect that model performance will further improve with larger datasets and by incorporating temporal dynamics of neural responses across more subjects.

We trained a convolutional neural net model based on the aggregated dataset to learn spatial representations of activated glomeruli. This architecture leverages local receptive fields to capture the spatial organization of regions of interest (ROIs) within the olfactory bulb. A schematic diagram and details of model is at Appendix F.

The table presenting the Precision, Recall and F1 score results on the test set for each odorant is placed at Appendix B. Figure 10 at Appendix C shows the confusion matrix, where each cell

shows the percentage of correct responses. Figure 11 at Appendix D shows per class and the Micro and Macro - average AUC $\approx 0.99$. The results demonstrate that a simple CNN trained on MPI images from mice can effectively separate neural representations by odor identity within the learned neural representation space of odors. We expect the accuracy of this simple model to decline as we increase the number of odorants, and particularly odorants that elicit similarly localized responses in the dorsal olfactory bulb that we observe. Indeed, we observe a few such cases of error in our sample.

The error patterns in this rudimentary projection of time-series data reveal multiple sources of potential "noise" in the data which can lead to the prediction errors shown in the confusion matrix in Appendix C Figure 10.Likely sources of error include: (a) inter-subject variability, (b) similarity of the ROIs for odors, and (c) experimental variance or observational error, such as fluctuations in image luminosity or spurious activity outside the olfactory bulb. Addressing these sources of noise will be critical for improving generalization as the dataset expands.

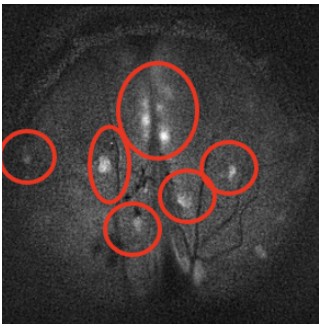 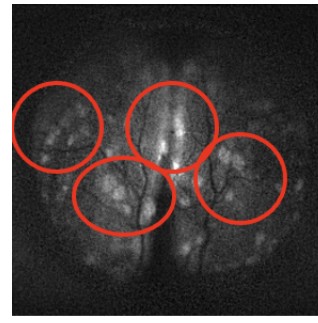

(a) Train Image - Pentyl Acetate             (b) Test Image - Pentyl Acetate

Figure 2: **Comparison of an Instance of Train and Test Image for Pentyl Acetate**. The red outlined areas help appreciate the inter-subject variability in ROIs for Pentyl Acetate.

Figure 2a and Figure 2b shows inter-subject variability in the activation regions for Pentyl Acetate (outlined in red). Figures 3a and 3b show a case of substantial overlap in MPI activation patterns for Methyl Benzoate and Benzaldehyde, resulting in 73% of Methyl Benzoate trials being misclassified as Benzaldehyde.

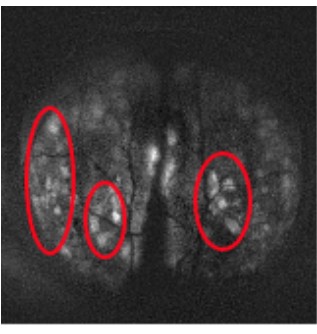 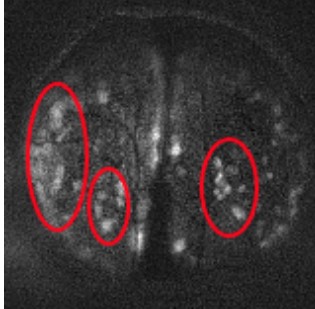

(a) Test Image - Methyl Benzoate             (b) Test Image - Benzaldehyde

Figure 3: **Comparison of an Instance of Test Image for Benzaldehyde and Methyl Benzoate**. The red outlined areas help appreciate similarity in ROIs for Methyl Benzoate and Benzaldehyde.

Although the two odorants share structural commonalities, including a benzene ring that contributes to their aromatic properties and may influence receptor interactions, their functional groups differ: Methyl Benzoate contains an ester, whereas Benzaldehyde contains an aldehyde. These differences could yield distinct binding affinities and downstream activation patterns across the olfactory bulb, but the overlap observed in our data highlights how structural similarity can drive misclassification.

We observe similar effects among short-chain carboxylic acids. Propionic Acid is frequently misclassified as Valeric Acid, Isobutyric Acid, and Acetic Acid, while Acetic Acid is commonly misclassified as Isobutyric Acid, Propionic Acid, and Valeric Acid. Additionally, we also identify misclassifications between odorants with divergent molecular structures, such as Geraniol and 2-Ethylhexanal. These cases emphasize that misclassification is not solely explained by gross structural similarity. Instead, they highlight the complex and nonlinear nature of odorant–receptor interactions that ultimately determine glomerular activation and neural representations.

We also observe instances of spurious activity outside the olfactory bulb. To examine how the model differentiates among odorant-specific activation patterns, we employ Gradient-weighted Class Activation Mapping (Grad-CAM) (Selvaraju et al., 2016).Grad-CAM highlights the spatial features most responsible for a given prediction, thereby localizing the regions that drive classification. Appendix E presents averaged MPI representations from both training and testing sets alongside the corresponding Grad-CAM heat maps (for each of the 35 odorants). These results reveal the localized activation patterns that the network leverages for classification. The grad cam visualizations employ a color-coded scheme to signify the model's prioritization within the bulb's regions: areas marked in red are deemed most significant by the model, followed by those in green. Blue zones are considered unimportant. Even at this aggregate level, we can see that the model can classify the odorants based on the activation patterns observed across various segments of the bulb for the odorants.

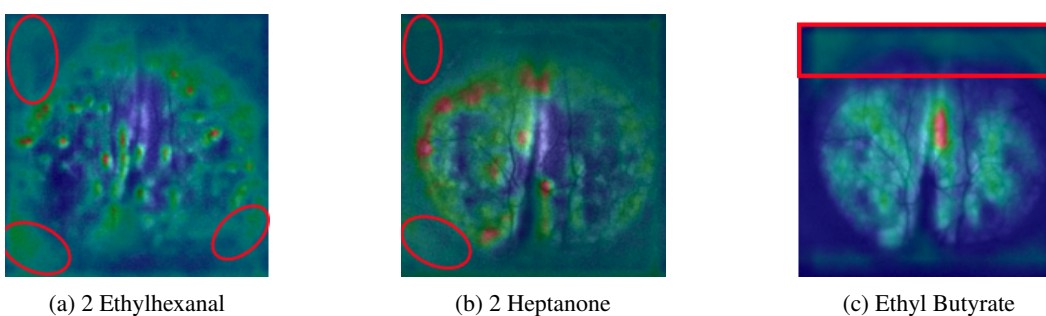

| (a) 2 Ethylhexanal | (b) 2 Heptanone | (c) Ethyl Butyrate |

Figure 4: **GradCAM Images Show Features Learnt by Model Outside Olfactory bulb**. The Grad CAM overlay images above show the discriminative regions considered important by the model.

Figure 4a, Figure 4b and Figure 4c show that the model considers features outside the olfactory bulb (outlined in red), which are clearly noise. This experimental error can be mitigated by preprocessing algorithms that focus exclusively on data within the bulb. Together with denoising, such algorithms are essential to attenuate background interference and noise in images to achieve better discriminative acuity.

## 5  DISCUSSION & NEXT STEPS

Our approach has clear scope for improvement, and we expect substantial gains with larger datasets and more expressive models. Neural networks trained on maximum-intensity-projection (MPI) images of glomerular activity learn spatial representations that support odor classification across mice. Grad-CAM visualizations indicate that the model leverages localized regions of interest (ROIs) consistent with glomerular patterns, though we also observe spurious attention outside the olfactory bulb. While Grad-CAM is useful for interpretability, it is model-dependent and sometimes highlights extra-bulbar regions. Removing vasculature and other background noise could further improve both predictive power and biological plausibility.

Beyond static MPI representations, incorporating the temporal sequence of glomerular activations is a critical next step. Prior work has shown that latency and order of activation carry significant information for odor perception (Chong et al., 2019). In particular, the primacy coding hypothesis suggests that a small set of the earliest-activated glomeruli can be sufficient to define odor identity (Wilson et al., 2017). Modeling this principle could yield more compact and interpretable representations, and help bridge biological coding theories with machine learning approaches to olfaction.

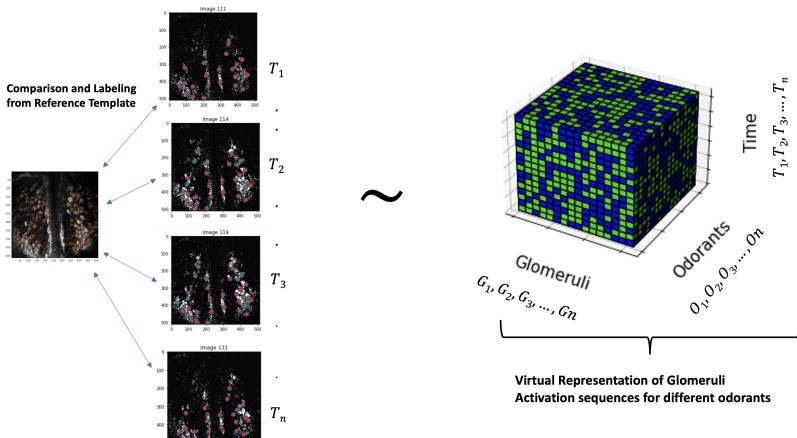

Figure 5: **A Virtual Model of Glomeruli Activation Sequence for different odorants**. The virtual representation shows comparison and labelling the Glomeruli activation for respective frames. The cube represents the Glomeruli, Odorants and Time on three different axes.

As our next step, we aim to develop models that jointly learn spatial and temporal representations of glomerular activity across a larger set of odorants. The objective is to localize ROIs consistently across mice and to construct a database representation as illustrated in Figure 5. In this representation, the $x$-axis indexes glomeruli, the $y$-axis encodes temporal activation sequences, and the $z$-axis corresponds to odorants. The resulting Expected Glomerular Activation Cube provides an archetypal spatiotemporal profile of odor-evoked responses that can serve as both a benchmark representation and a target for predictive modeling. Looking forward, this can serve as the output label in generative frameworks, where models predict activation tensors conditioned on molecular or linguistic descriptors of odorants. Given the high dimensionality and nonlinearity of odor space, achieving this goal will require substantially larger datasets. To this end, we are expanding Odor-MNIST (oMNIST) toward a thousand-odor benchmark, with data organized along the lines of Figures 5 and 6.The dataset with trial videos for 35 odorants has been made publicly available.[2]

## 6 CONCLUSION

In summary, our position is that there is no simple way to parametrize the space of molecules into an "odor map" at the moment, and little is known about the geometry of olfaction, such as the relationship between individual molecules and mixtures that contain them. This approach has translational potential in diagnostics, as many diseases are challenging to detect with conventional clinical methods and yield VOC profiles that can be subtle even for GC–MS, yet they are often detectable by trained animals. Models of glomerular activation may offer a path to capture these biologically relevant signatures and translate them into scalable diagnostic tools.

Artificial intelligence has advanced rapidly in language and vision through abundant datasets and algorithms that learn representation from data. Olfaction is now becoming accessible to similar approaches thanks to improved imaging of pre-synaptic activity at the first olfactory synapse. Our results show that neural networks trained on maximum-intensity-projection (MPI) images of glomerular activity can learn spatial representations that support accurate, cross-animal odor classification. These learned embeddings offer a data-driven coordinate system for studying odor identity at the level of neural signals. Machine learning methods will therefore be invaluable for solving open questions in olfaction, especially if this approach considers the latent biological variables involved, that shed light on the central role of odorant-receptor interactions in smell(Barwich & Lloyd, 2022). Accordingly, our primary goal is to advance the science of olfaction through a Machine Learning approach which is embedded in biology.

---

[2]Dataset link omitted for double-blind review. It will be released after review is completed. Processed data alongwith the code is available at  Github Repository

**Ethics Statement**. All animal procedures were approved and conducted as per IACUC Protocol #IA16-00197 and complied with relevant guidelines.

**Reproducibility Statement**. The pre-processed data along with the code is available at Github Repository. The raw video data has been publicly released. The link for raw data is omitted only for double blind review to avoid disclosing author identities. The link will be released upon completion of review.

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

# A  BUILDING THE DATA INGESTION PIPELINE

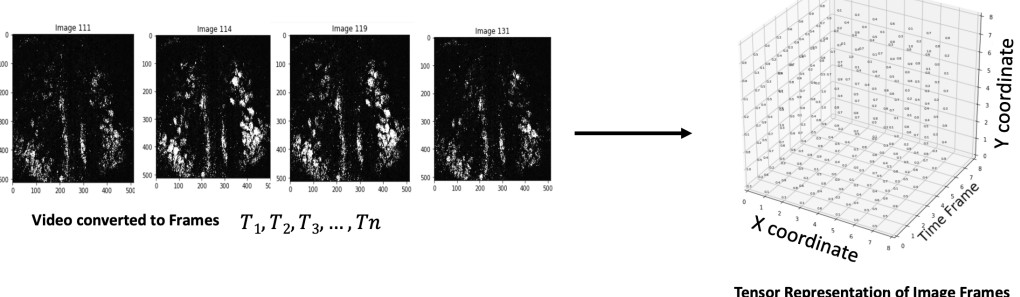

Figure 6: **Curating the Videos into Image Frames.** The diagram above is a schematic representation of the curation of image frames from the glomeruli activation sequence video. The image stack is converted into a 3d tensor representation. The $x$, $y$ and $z$ axis represent the x, y coordinates of ROIs and image frames over the time period of the video respectively.

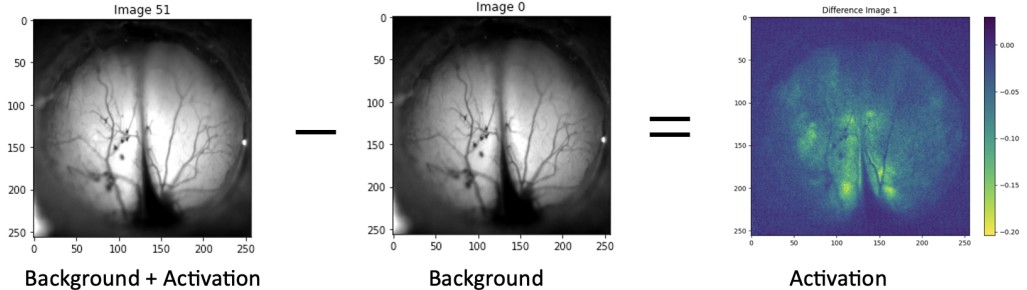

Figure 7: Background Subtraction during Noise Removal from Raw Image Frames.

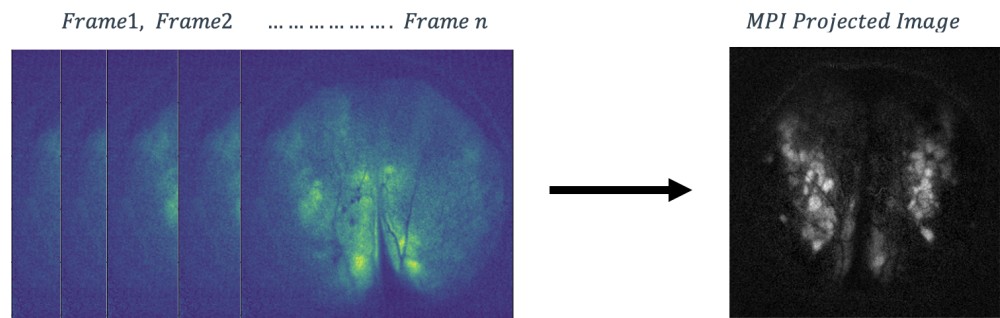

Figure 8: Image Frames of Glomeruli Activation Video converted to a single MPI projected image.

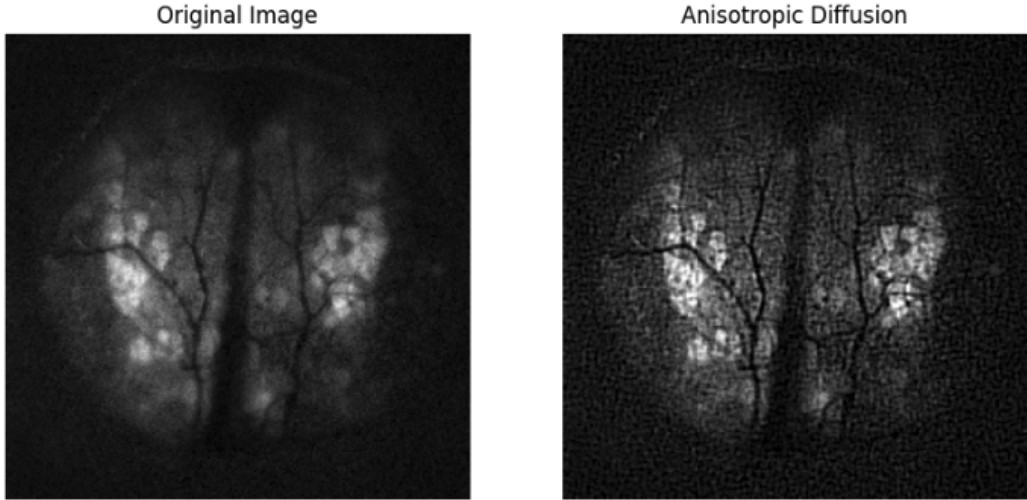

Figure 9: Denoising of MPIs using Anisotropic Diffusion.

# B PRECISION, RECALL AND F1 RESULTS ON TEST DATA

Table 1: **Results on Test Data**

| S No. | Odorant | Precision | Recall | F1 |
|---|---|---|---|---|
| 1. | 2 3 Pentanedione | 0.80 | 0.80 | 0.80 |
| 2. | 2 Ethylbutyric Acid | 0.50 | 0.45 | 0.48 |
| 3. | 2 Methyl Butyraldehyde | 1.00 | 1.00 | 1.00 |
| 4. | 2 Methyl Valeraldehyde | 0.85 | 1.00 | 0.92 |
| 5. | 2 Methylhexanoic Acid | 0.48 | 1.00 | 0.65 |
| 6. | 2 Ethylhexanal | 0.57 | 0.40 | 0.47 |
| 7. | 2 Heptanone | 1.00 | 0.64 | 0.78 |
| 8. | 33 Dimethylbutyric Acid | 1.00 | 1.00 | 1.00 |
| 9. | 3 Methylvaleric Acid | 0.90 | 0.82 | 0.86 |
| 10. | 3 Heptanone | 0.56 | 0.91 | 0.69 |
| 11. | 4 Methylvaleric Acid | 0.53 | 0.80 | 0.64 |
| 12. | 5 Methyl 2 Hexanone | 0.45 | 0.90 | 0.60 |
| 13. | Acetic Acid | 0.17 | 0.08 | 0.11 |
| 14. | Butyl Acetate | 0.38 | 0.55 | 0.44 |
| 15. | Cyclopentane Carboxylic Acid | 1.00 | 0.77 | 0.87 |
| 16. | Ethyl Tiglate(ET) | 1.00 | 1.00 | 1.00 |
| 17. | Ethyl Butyrate | 0.40 | 0.73 | 0.52 |
| 18. | Heptanoic Acid | 1.00 | 0.18 | 0.31 |
| 19. | Methyl Valerate (MVT) | 0.57 | 0.73 | 0.64 |
| 20. | Methyl Benzoate | 0.33 | 0.09 | 0.14 |
| 21. | Pentyl Acetate | 0.80 | 0.36 | 0.50 |
| 22. | Salicyl Aldehyde | 0.83 | 1.00 | 0.91 |
| 23. | Benzaldehyde | 0.78 | 0.64 | 0.70 |
| 24. | Butyraldehyde | 1.00 | 0.45 | 0.62 |
| 25. | Butyric Acid | 0.75 | 0.67 | 0.71 |
| 26. | Cinnamaldehyde | 1.00 | 1.00 | 1.00 |
| 27. | Ethyl Valerate | 1.00 | 0.55 | 0.71 |
| 28. | Geraniol | 0.86 | 0.55 | 0.67 |
| 29. | Heptyl Acetate | 0.60 | 0.27 | 0.37 |
| 30. | Isobutyric Acid | 0.53 | 0.73 | 0.62 |
| 31. | M Anisaldehyde | 1.00 | 1.00 | 1.00 |
| 32. | N Methylpiperidine | 1.00 | 0.94 | 0.97 |
| 33. | P Anisaldehyde | 0.79 | 1.00 | 0.88 |
| 34. | Propionic Acid | 0.27 | 0.27 | 0.27 |
| 35. | Valeric Acid | 0.14 | 0.20 | 0.17 |

# C  CONFUSION MATRIX FOR ODOR CLASSIFICATION

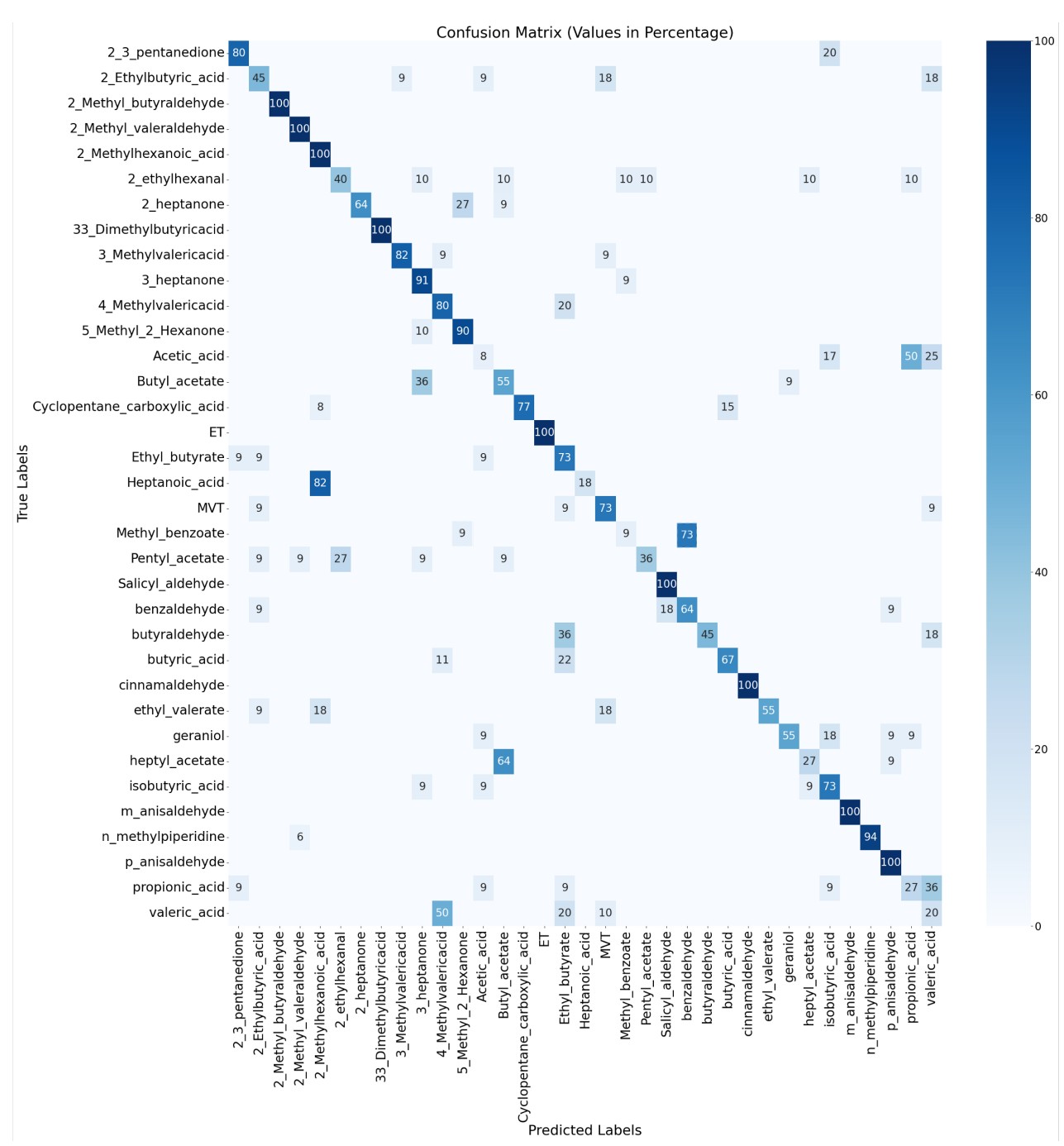

Figure 10: Confusion Matrix for classification of Thirty Five Odors based on Neural Representation of Activated Glomeruli in Olfactory bulb of Test Mice

# D   ROC CURVE ACROSS 35 ODORS

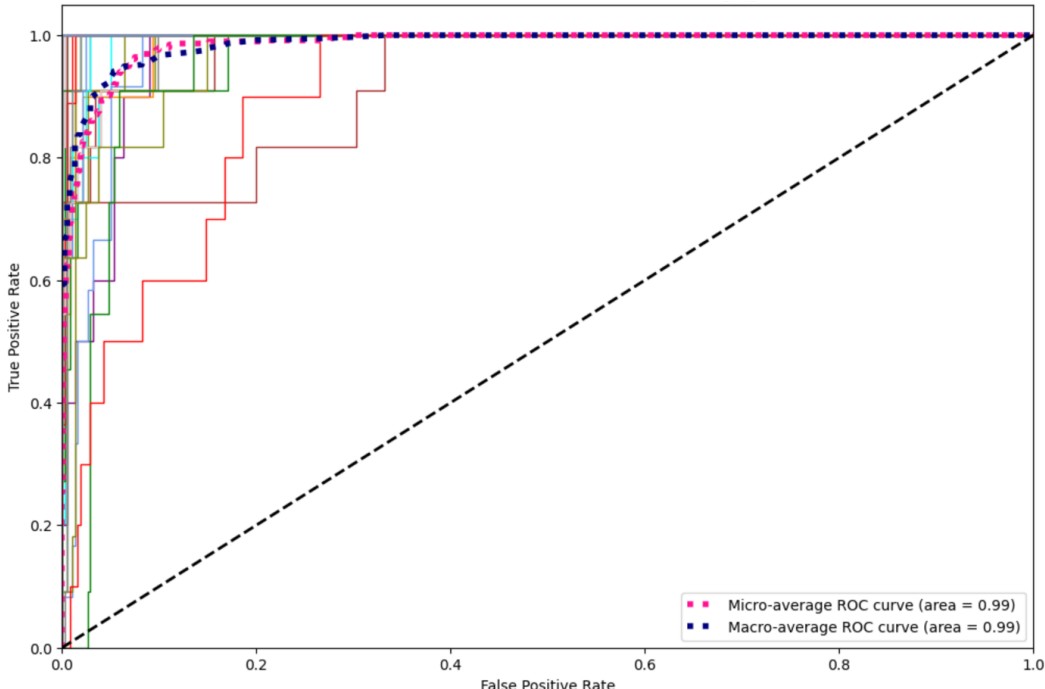

Figure 11: **One-vs-rest ROC curves across 35 odorants**. Thin lines show per-class ROCs, while the dotted magenta and dashed blue curves denote micro- and macro-averages, respectively (AUC = 0.99 for both). The diagonal line indicates chance level.

*The above Figure shows ROC curves under a one-vs-rest evaluation. The micro-average (dotted magenta) aggregates predictions across all classes and therefore reflects class imbalance, while the macro-average (dashed blue) gives equal weight to each odorant by averaging per-class ROCs. Both scores are high (AUC  0.99), showing that the model generally ranks the true class above alternatives. However, AUROC is threshold-independent and dominated by many easy negatives, so it may remain high even when top-1 classification accuracy is modest for confusable odorants. To address this, we also report per-class precision, recall, and F1 scores as complementary measures.*

# E    GRAD CAM OVERLAY ON TEST IMAGES

| Figure No. | Odorant Name | Avg Training Mouse Image | Avg Test Mouse Image | Grad CAM Overlay Image |
|---|---|---|---|---|
| 1. | 2 3 Pentanedione | | | |
| 2. | 2 Ethylbutyric Acid | | | |
| 3. | 2 Methyl Butyraldehyde | | | |
| 4. | 2 Methyl Valeraldehyde | | | |
| 5. | 2 Methylhexanoic Acid | | | |
| 6 | 2 Ethylhexanal | | | |
| 7. | 2 Heptanone | | | |
| 8. | 33 Dimethylbutyric Acid | | | |
| 9. | 3 Methylvaleric Acid | | | |

The Grad CAM overlay images above show the discriminative regions considered important by the model. Areas marked in red are deemed most significant by the model, followed by those in green. Blue zones are considered unimportant. The above mapping shows the correctly classified images.

| Figure No. | Odorant Name | Avg Training Mouse Image | Avg Test Mouse Image | Grad CAM Overlay Image |
|---|---|---|---|---|
| 10. | 3 Heptanone | | | |
| 11. | 4 Methylvaleric Acid | | | |
| 12. | 5 Methyl 2 Hexanone | | | |
| 13. | Acetic Acid | | | |
| 14. | Butyl Acetate | | | |
| 15. | Cyclopentane Carboxylic Acid | | | |
| 16. | Ethyl Tiglate(ET) | | | |
| 17. | Ethyl Butyrate | | | |
| 18. | Heptanoic Acid | | | |

The Grad CAM overlay images above show the discriminative regions considered important by the model. Areas marked in red are deemed most significant by the model, followed by those in green. Blue zones are considered unimportant. The above mapping shows the correctly classified images

| Figure No. | Odorant Name | Avg Training Mouse Image | Avg Test Mouse Image | Grad CAM Overlay Image |
|---|---|---|---|---|
| 19. | Methyl Valerate (MVT) | | | |
| 20. | Methyl Benzoate | | | |
| 21. | Pentyl Acetate | | | |
| 22. | Salicyl Aldehyde | | | |
| 23. | Benzaldehyde | | | |
| 24. | Butyraldehyde | | | |
| 25. | Butyric Acid | | | |
| 26. | Cinnamaldehyde | | | |
| 27. | Ethyl Valerate | | | |

The Grad CAM overlay images above show the discriminative regions considered important by the model. Areas marked in red are deemed most significant by the model, followed by those in green. Blue zones are considered unimportant. The above mapping shows the correctly classified images.

| Figure No. | Odorant Name | Avg Training Mouse Image | Avg Test Mouse Image | Grad CAM Overlay Image |
|---|---|---|---|---|
| 28. | Geraniol |  |  |  |
| 29. | Heptyl Acetate |  |  |  |
| 30. | Isobutyric Acid |  |  |  |
| 31. | M Anisaldehyde |  |  |  |
| 32. | N Methylpiperidine |  |  |  |
| 33. | P Anisaldehyde |  |  |  |
| 34. | Propionic Acid |  |  |  |
| 35. | Valeric Acid |  |  |  |

The Grad CAM overlay images above show the discriminative regions considered important by the model. Areas marked in red are deemed most significant by the model, followed by those in green. Blue zones are considered unimportant. The above mapping shows the correctly classified images.

## F    DETAILS OF CNN MODEL ARCHITECTURE

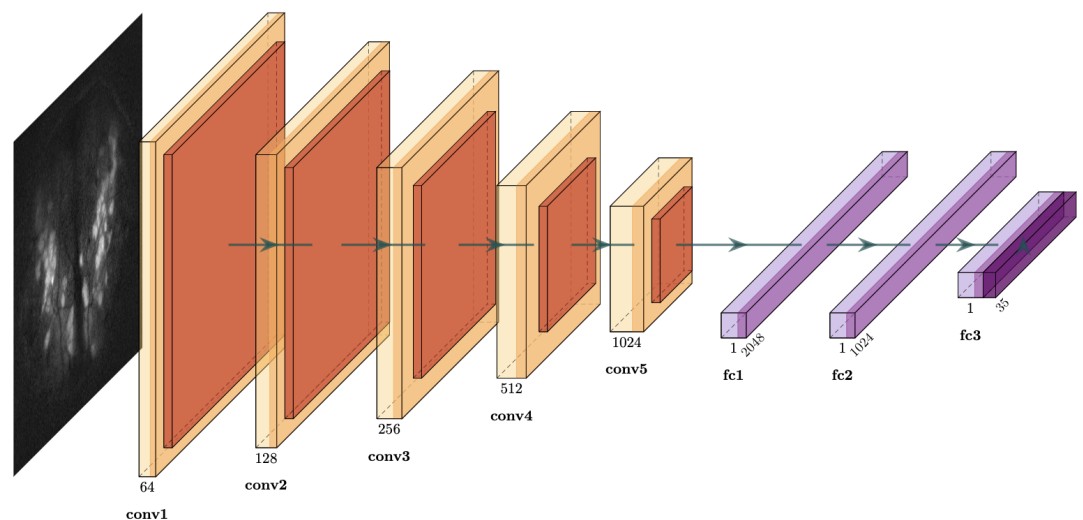

Figure 12: **Schematic Diagram of CNN Architecture**. The diagram above is a schematic representation of the CNN used for classification of MPI images. The Layers in brown indicate the convolutional and the pooling layers. The layers in magenta are fully connected layers with a softmax layer at the end.

The convolutional neural net used for the classification comprises five convolutional layers that take a single-channel input and applies 64 filters with a kernel size of 3x3, using a stride of 1 and padding of 1 to preserve the spatial dimensions of the input. We increase the number of filters, doubling from one layer to the next: conv2 has 128 filters, conv3 has 256 filters, conv4 has 512 filters, and conv5 has 1024 filters. Each of these layers also uses 3x3 kernels with a stride of 1 and padding of 1, enabling the network to learn increasingly complex and abstract features at each layer. Following each convolutional layer, a batch normalization layer is applied to normalize the output of the convolutional layers, reducing internal covariate shift and stabilizing the learning process. We use max pooling layer to downsample the feature maps. These high level features learned by the CNN are flattened and passed through three fully connected layers. The final fully connected layer maps the 1024-dimensional features to 35 output classes. For this multi-class classification problem, we minimize a standard cross entropy loss function and choose the class with the highest predicted probability. We used a dropout rate of 0.3 to prevent overfitting.

