# OpenReview forum: "Learning Representations from Pre-synaptic Glomerular Responses for Odor Classification"
_ICLR.cc/2026/Conference — Submitted to ICLR 2026_

### Official Review · Reviewer_qzrb · 2025-10-30

**Soundness:** 2
**Presentation:** 2
**Contribution:** 1
**Rating:** 2
**Confidence:** 4

**Summary:**

The paper addresses te problem of how odor identity can be decoded from neural representations at the first synaptic stage of the olfactory system. A parallel to visual sensing is done and simple methodology deployed to study if an how machine learning can be used to  learn meaningful representations from calcium imaging of glomerular activations.

**Strengths:**

The positive aspect of the paper is its pedagogical description of olfactory modelling complexity and the parallel made to visual sensing. Paper is clear and easy to read.

**Weaknesses:**

Despite its clearness, the paper lacks original contribution. There is no new methodology developed and the insights given the generated dataset seems to preliminary to assess the validity of the presented results. The authors also claim that there are sources of noise and that addressing these  will be critical for improving generalization as the dataset expands in the future.

**Questions:**

This is a rather preliminary study and it is important to see more evidence on how the expended dataset would perform given the complexity of the odor space. I support the idea of biological approach but there are many ongoing chemical and psychological studies that involve human subjects using EMG, EEG and alike and the paper would benefit from a deeper discussion of the approach with respect to the state of the art.

---

### Official Review · Reviewer_k4c6 · 2025-10-31

**Soundness:** 2
**Presentation:** 2
**Contribution:** 2
**Rating:** 2
**Confidence:** 4

**Summary:**

In this paper, the Authors collect a dataset of glomerular activations and train a convolutional neural network model to decode the odorants that have evoked the glomerular activation patterns. They evaluate the prediction quality of the model, analyze the sources of the confusions, and use attribution techniques to determine the basis of the model's classification decisions. The work concludes that the odor identity decoding is possible rom time-averaged images of glomerular activity and that this result generalizes to new mice.

**Strengths:**

I think the approach taken here by the Authors and the dataset collected in this work are highly valuable. As the Authors mention in the paper, olfaction is a complex nonlinear process with limited data. Thus, breaking this process into stages and studying each stage separately is a higly reasonable choice. This way, the glomerular activation dataset allows isolating the neural processing of odorants (separated from the mass transport problems etc).

The CNN model trained by the Authors is useful as a proof of concept: It shows that the data provided in the dataset is sufficient to successfully decode the odor identity and to generalize to new subjects.

The paper contains text that introduces olfaction and the problems it faces to a machine learning reader, who might not be familiar with the field.

**Weaknesses:**

Primarily, I am not sure about what's the main stated contribution of the paper. Is this the dataset of the glomerular activity? Is this the neural network model that can decode the odorant identity from the glomerular data? If so, is the primary novelty in the ability to decode from the levels of activity or is the main stated novelty in decoding from raw videos? Or does the neural network serve as a proof of concept that the data in the dataset contains the relevant signal? Or is it in the interpretability analysis? Currently it looks like a paper touches upon all these things but doesn't explore any of them in depth, as follows. A Dataset paper would be generally expected to provide the results of several baseline models to eastablish that the task is neuither too easy nor too hard. A machine-learning model paper would be generally expected to evaluate multiple design choices for the model and to show abalations determining whether these choices are beneficial for the model. A basic-science paper would have a clear-cut biological result: An ability to decode from glomerular activations is known, so I'm not sure what such a result would be here.

Secondly, the results (both in the main text and in the Appendix) miss details. Generally, it is expected that, from a description in the text here, one would be able to reproduce the work. While the description of the data mentions GCaMP, it is unclear how the data was collected and imaged including the ways the odorants were administered to the animals (which, as you know, is extremely important in olfaction). While the description of the preprocessing lists the filters and some additional details may be inferred from the labels in supplementary figures, the specific parameters for the filters are missing, making the preprocessing not reproducible. While the description of the model provides the model's architecture, the actuual training parameters are not provided either.

Lastly, most of the paper is spent on describing color vision (which is typical for papers in olfaction). However, due to the limited page count, this has left too little space for the results. Even though I have read the appendinx, neither the reviewers nor the readers are expected to do it in this venue. The appendix is supposed to be a place for minor additional details, not the main results and methods.

Minor:

- Unclear if the images were stabilized (registered) over time to compensate for the motion artifacts

- The reason to average the images over time is unclear: The temporal dynamics of glomerular activations may carry the information relevant to the olfactory code.

- The reason to use CNNs (as opposed to identifying the glomeruly as ROIs and treating their activity as time series) is unclear

- Unclear why the model is expected to genberalize, provided that the locations of the glomeruly may somewhat vary accross mice.

It would be exciting to see a revision of this paper where 1) the scope is clearly defines; 2) the method is detailed; 3) the design choices are compared to alternatives both on the ML side and on the biological hypothese side; 4) the paper is formatted in line with the venue's specifications. Until at least some of these aspects are fulfilled, I, sadly, find the manuscript not ready for the publication here.

**Questions:**

N/A

---

### Official Review · Reviewer_BD4W · 2025-11-01

**Soundness:** 1
**Presentation:** 1
**Contribution:** 2
**Rating:** 2
**Confidence:** 4

**Summary:**

The authors propose oMNIST, a series of images corresponding to glomerular activations in response to olfactory stimuli. 35 monomolecular odorants are provided to mice, and the corresponding brain signals are recorded using a camera as a video sequence. The authors provide some analysis by training a CNN on oMNIST. This work shows a lot of promise and seeks to verify a key hypothesis by linking biological signals to olfactory perception, but in its current form, the work performed is very preliminary, with their novel results significantly underevaluated. Therefore, I recommend a rejection of this paper from ICLR.

**Strengths:**

The problem the authors are attempting to tackle is very novel and interesting. I appreciate the provision of large and curated public datasets for olfaction, given the fact that it has traditionally been quite a data-scarce domain. At its full potential, with a large number of olfactants investigated at multiple concentrations across multiple subjects for reproducibility, oMNIST could truly be useful in understanding olfactory phenomenon.

**Weaknesses:**

The introduction contains almost no citations and I am disheartened to see this. Almost every sentence within the introduction likely draws upon conclusions arrived at in prior work that the authors fail to cite. Parts of the introduction remain vague because of this, e.g. “Smell… remains poorly understood as a sense for a number of reasons” -> what reasons? In other parts of the work, I cannot tell if the authors are proposing original claims, or if they are summarizing things that are already understood. The difficulty of olfaction as a problem for machine learning has already been discussed in a series of prior works (e.g. Lee at al. 2023, 10.1126/science.ade4401, which the authors cite), so I’m not sure what the authors’ original propositions are in this piece of work. In other parts of the work, the authors refer to “recent advances in optical methods” but do not provide a source for this claim.

The comparisons between vision and olfaction, and many of the justifications provided for their work, are approached at a very basic level. I truly appreciate the authors’ breaking down the concepts within the paper, but I believe that in pursuing this analysis, the authors have significantly underevaluated their dataset and key results to meet the page limit. Many details (e.g. choosing mice vs humans, then evaluating the limitations of cell lines) could be shifted to the Appendix as it does not impact the evaluation of the results the authors discuss. Conversely, key results like Figure 7 and Appendices B, C and D should have been in the main text. Reviewers are not required to read the appendices and one can interpret that the authors have not deemed the model performance important enough to place in the main text.

I’m not sure who the intended audience is for this piece of work, but it feels like the extent of which these topics need to be described suggest that this conference might not be the right avenue for this piece of work. To me, it sounds like the authors are olfactory scientists attempting to instruct optical scientists on the difficulty of olfaction. Perhaps a weak link can be drawn between optics and computer vision, but the work mostly focuses on human/mouse perception instead. The authors also spend ~5 pages motivating the comparison between vision and olfaction, but never revisit it again in the discussion of the results, so I am left wondering what the point of discussing all that was.

Condensing the temporal information heavily compresses the information for the glomerular activations, and there are many temporal architectures that can exploit this level of detail (e.g. transformers). The authors recognize that the temporal dynamics is important -- but this should have been the baseline to begin with, unless there was a good prior to condense the video into a single image.  The comparisons of individual images in Figures 2 and 3 are also performed on a qualitative level and instead the comparison should be done using a quantitative image-based difference between the two ground truth images -- and compare it against what the learned representational distances are.

The work seems very preliminary. I don’t think it’s a well-established fact that representations can be learned from glomerular responses and used for classification, and even though the authors set out to investigate this hypothesis, they did not succeed in doing so. As the authors evaluate in Sec 4.3, there are significant numbers of misclassifications that occur, and I can’t pinpoint at which part of the machine learning pipeline the error occurs -- whether it’s overcompression in data preprocessing or that the hypothesis is not valid to begin with. The authors provide some analysis on where they think the errors occur between Figures 2-4, but in general, I think that it is hard to arrive at anything conclusive or general about the quality or utility of the data as we only have a comparison between training on one mouse and predicting on the other.

There is also a multitude of typographical errors:
- Line 128: “highly highly”
- Line 129: “thaht”
- Line 214: “to a read out semantic descriptors”
- Line 215: missing the AUROC abbreviation

**Questions:**

1) Could you please highlight to me what claims in your paper are original, and which ones are based on prior literature, by citing the appropriate sources?
2) Can you consider a temporal baseline to report your model performance?
3) There are many key things about the experimental protocol that are missing. What concentrations are provided for each stimuli? How many concentrations are collected per stimuli? Are stimuli provided multiple times to each mouse? How is the stimuli introduced to the mouse? Is there any control for the environment that the mouse receives the stimulus? How many of such images do you have for each identity-response pair?
4) Could you provide quantitative metrics for analyzing the image differences in brain activation pattern? What about metrics for molecular structure differences to help guide the discussion in misclassification? Is the relationship between molecular structure and glomerular response consistent throughout the dataset other than the specific examples that you’ve highlighted between lines 374-384?

---

### Meta-Review · Area_Chair_bGGG · 2026-01-04

**Summary:**

This submission has the potential to be an important contribution to olfactory science (and the application of machine learning to olfaction); however, the reviewers expressed concerns about the current state of the contribution.

1. The reviewers expressed concern about the framing of the paper:

> To me, it sounds like the authors are olfactory scientists attempting to instruct optical scientists on the difficulty of olfaction (Reviewer BD4W)

> Primarily, I am not sure about what's the main stated contribution of the paper (Reviewer k4c6)

2. The work also appears incomplete:

> The work seems very preliminary (Reviewer BD4W)

> Secondly, the results (both in the main text and in the Appendix) miss details (Reviewer k4c6)

**Reviewer Concerns:**

None of the reviewers concerns were addressed in the rebuttal (the authors did not rebut).

**Reviewer Scores:**

Since the authors did not responds to the reviewer comments, I do not think that the reviewers would have changed their scores.

---

### Decision · Program_Chairs · 2026-01-26

Reject